# Pre-Existing HSV-1 Immunity Enhances Anticancer Efficacy of a Novel Immune-Stimulating Oncolytic Virus

**DOI:** 10.3390/v14112327

**Published:** 2022-10-23

**Authors:** Jun Ding, Yanal M. Murad, Yi Sun, I-Fang Lee, Ismael Samudio, Xiaohu Liu, William Wei-Guo Jia, Ronghua Zhao

**Affiliations:** 1Virogin Biotech Canada Ltd., Vancouver, BC V6V 3A4, Canada; 2Shanghai Virogin Biotech Ltd., Shanghai 200240, China; 3Immunity Diagnostics Inc., Vancouver, BC V5Z 1E3, Canada; 4CNBG-Virogin Biotech (Shanghai) Ltd., Shanghai 200240, China

**Keywords:** oncolytic viruses, immunotherapy, HSV-1

## Abstract

Oncolytic viruses (OVs) can specifically replicate in the host and cause cancer cell lysis while inducing an antitumor immune response. The aim of this study is to investigate the impact of either pre-existing immunity against herpes simplex virus type-1 (HSV-1) or multicycle treatment with OVs on anticancer efficacy of VG161, an HSV-1 OV in phase 2 clinical trial. VG161 efficacy was tested in CT26 mouse models by comparing the efficacy and immune response in naïve mice or in mice that were immunized with VG161. Moreover, VG161 efficacy in HLA-matched CD34+ humanized intrahepatic cholangiocarcinoma (ICC) patient-derived xenograft (PDX) models was also tested in multicycle treatment and was compared to standard chemotherapy for this type of cancer (gemcitabine). The HSV-1-immunized mice significantly inhibited tumor growth in VG161-treated mice compared to control naïve treated mice. RNA expression profiling and ELISPOT analyses indicated changes in the tumor’s immune profile in the immunized and treated group compared to naïve and treated mice, as well as enhanced T cell function depicted by higher numbers of tumor specific lymphocytes, which was enhanced by immunization. In the ICC PDX model, repeated treatment of VG161 with 2 or 3 cycles seemed to increase the anticancer efficacy of VG161. In conclusion, the anticancer efficacy of VG161 can be enhanced by pre-immunization with HSV-1 and multicycle administration when the virus is given intratumorally, indicating that pre-existing antiviral immunity might enhance OV-induced antitumor immunity. Our results suggest potential clinical benefits of HSV-1-based OV therapy in HSV-1-seropositive patients and multicycle administration of VG161 for long-term maintenance treatment.

## 1. Introduction

Oncolytic viruses (OVs) are diverse natural or genetically modified viruses that can infect, specifically replicate in, and thereby cause lysis of tumor cells. With the release of tumor antigens from the lysed tumor cells and the activation of anticancer immunity induced by the virus itself or together with the payloads expressed by the virus, OVs may overcome immune suppression in the tumor microenvironment and trigger systemic specific antitumor immune responses after intratumoral (IT) injection [1]. A diverse range of OVs have shown efficacy in preclinical studies (reviewed in [2,3]). However, only three OVs have cleared the hurdle of regulatory approval—adenovirus H101, approved in China in 2005 to treat head and neck cancer [4], Talimogene laherparepvec (T-VEC), approved by the FDA in 2015 for the treatment of advanced melanoma [5], and in 2021, Delytact^®^, an HSV-1-based OV (G47∆; teserpaturev) was granted conditional approval in Japan for the treatment of glioblastoma (GBM) [6]. It has become clear that direct infection and lysis of tumor cells is often not sufficient to generate a significant antitumor response. T-VEC was optimized for immunotherapy by expressing the cytokine granulocyte-macrophage colony-stimulating factor (GM-CSF) and can generate systemic antitumor immunity as evidenced by observations of tumor regression in non-injected lesions [4,7,8,9,10]. However, the overall durable response rate in patients treated with T-VEC is below 20% [8,10]. Therefore, it is apparent that a more robust immune response must be elicited in addition to potent oncolysis for an OV therapy to achieve long-lasting efficacy in the clinic.

VG161 is a novel type 1 herpes simplex virus (HSV-1) OV with deletion of the neurovirulence gene ICP34.5; it is capable of delivering four immunomodulatory molecules into the tumor microenvironment—IL-12, IL-15, IL-15 receptor alpha subunit isoform 1 (IL-15RA), and a fusion protein (TF-Fc) capable of blocking PD-1/PD-L1 interactions [11]. It replicates efficiently and exhibits robust cytotoxicity in multiple tumor cell lines. Moreover, the encoded cytokines (IL-12, IL-15/IL-15RA) and TF-Fc work cooperatively to boost immune cell function. In vivo testing in syngeneic CT26 and A20 tumor models reveals superior efficacy when compared to a backbone virus that does not express exogenous genes [11]. IT injection of VG161 induces abscopal responses in non-injected distal tumors and grants resistance to tumor re-challenge. The robust antitumor effect of VG161 is associated with T cell and NK cell tumor infiltration, expression of Th1-associated genes in the injection site, and increased frequency of splenic tumor-specific T cells. In addition, VG161 displayed a superb safety profile in GLP acute and repeated injection toxicity studies performed using cynomolgus monkeys and has entered phase I clinical trials [12].

HSV-1 seropositivity prevalence has been reported to be as high as 90% of the worldwide population [13,14]. Our preclinical studies also demonstrated that the intramuscular injection of VG161 can induce the anti-HSV-1 antibody. Therefore, it is critical to understand the potential impact of pre-existing anti-HSV-1 immunity on the efficacy of IT injection of VG161. The aim of this study is to investigate the impact of pre-existing immunity against HSV-1 on anticancer efficacy of VG161 and the related underlying mechanisms.

## 2. Materials and Methods

### 2.1. Investigational Product (IP) VG161

VG161 is an attenuated HSV-1 OV expressing human IL-12, IL-15, IL-15 receptor alpha subunit isoform 1 (IL-15RA), and secretable PD-L1 blocking peptide TF conjugated to human IgG4 Fc (TF-Fc). The construction and structure of VG161 has been reported by Chouljenko et al. [11]. To facilitate in vivo testing in a variety of mouse models, mVG161 was constructed; it is identical to VG161 apart from mouse IL-12 replacing human IL-12 and the presence of a mouse-specific version of the PD-L1 blocker peptide TF conjugated with mouse IgG1 Fc. Human IL-15 was retained in mVG161 due to its cross-reactivity with mouse cells [15,16,17,18]. The OV used in this study is a pilot-scale product manufactured by Genesail Biotech (Shanghai) under GMP conditions. The concentration of VG161 was >1 × 10^7^ PFU/mL, 1 mL/vial. The OV product was stored at −80 °C. A mouse version of VG161 (labeled as mVG161) expressing mouse IL-12, human IL-15, IL-15RA, and mouse TF-Fc was manufactured by Shanghai Virogin under GLP-like conditions and used as an immune-competent mouse model.

### 2.2. Animal Models and Administration of mVG161

CT26 immune-competent tumor-bearing mouse models: All experimental animal procedures were approved by the Animal Care Committee Guangzhou Curegenix Biotech company. Balb/c mice (5 weeks) were purchased from Beijing Vital River Laboratory Animal Technology Co., Ltd. and divided into two groups after quarantine. The first group was subcutaneously immunized with mVG161 at 5 × 10^5^ PFU/mouse on day 1 followed by a boost (additional dose level) on day 7. The second group was not immunized. On day 7 post boost, the blood was collected by tail nick, then the serum titer of anti-HSV-1 antibodies was determined by a commercially available kit following instructions from the manufacturer (Mouse/Rat HSV-1 IgG ELISA kit, Calbiotech, El Cajon, CA, USA. Catalog #H1029G-100). 

CT26 tumor cells were purchased from ATCC (CRL-2638). Cells were cultured in Dulbecco’s Modified Eagle Medium (DMEM) with 10% fetal bovine serum (FBS) and harvested using 0.25% trypsin while in exponential growth phase (~80–90% confluence). Cells were resuspended in DMEM with 10% FBS prior to cell counting, then centrifuged at 225× *g* for 10 min at 4 °C. Cell pellets were resuspended in phosphate-buffered saline (PBS) to a concentration of 1 million cells/100 µL and stored on ice. Tumor cells were subcutaneously implanted into the lower right and left flanks (for the bilateral tumor model) of each mouse (1 × 10^6^ cells/site) day 7 post boost. Once the tumors reached mean 100 mm^3^ in size, mice from each group were intratumorally injected with either mVG161 at a dose of 5 × 10^6^ PFU/mouse or vehicle, once a day for 3 days. For the bilateral tumor model, only one of the two tumors was injected (injected tumor), and the other (non-injected tumor) was observed. Observation of tumor growth was conducted every day after implantation, and tumor volume (TV) was measured using a digital caliper 3 times/week and calculated using the following formula: Tumor Volume = ½ × a × b^2^, (a = longest diameter in mm, b = shortest diameter in mm). T/C ratio (treatment-to-control ratio) was calculated using the following formula: T/C% = 100% × RTV treatment/RTV control (RTV = V_t_/V_0_) (V_t_ = tumor volume on measurement day, V_0_ = tumor volume one day before the virus treatment, RTV = residual tumor volume).

### 2.3. Patient-Derived Xenograft (PDX) Humanized Mouse Models

All experimental animal procedures were approved by the Animal Care Committee of Beijing IDMO Biotech company. PDX tumor tissue was collected from tumor specimens of patients with intrahepatic cholangiocarcinoma (ICC), dissected into pieces of 3 × 3 × 3 mm, then subcutaneously implanted into humanized Hu-NPI mice (35 W, female) established by transplantation of 1 × 10^4^ human CD34 stem cells with the HLA matched with PDX tested by PCR. When the tumor volume reached 100–150 mm^3^, animals were divided into four groups with 10 mice in each group: negative control (vehicle), positive control (combination of chemotherapeutic agents oxaliplatin 5 mg/kg and gemcitabine 100 mg/kg; OX + GEM), and two VG161-treated groups (VG161-H: high dose 7.75 × 10^6^ PFU/mouse and VG161-L: low dose 3.10 × 10^5^ PFU/mouse); the treatments were given in three cycles. For each cycle, the animals received IT injection of VG161 once per day for 5 days on days 1–5, 29–33, and 57–61. Animal body weight and tumor volume were measured as described above. T/C% was also calculated.

### 2.4. ELISPOT Assay to Verify Specific Anticancer Immunity

Mouse interferon (IFN)-γ ELISPOT assays (Mabtech, Cincinnati, OH, USA) were performed according to the manufacturer’s instructions. Briefly, splenocytes isolated from treated mice and control mice (10 mice per treatment condition) were added to each well of a 96-well plate (100,000 cells/well) and stimulated overnight with CT26 cells (5000 cells/well) to detect CT26-specific responses. Results were expressed as the number of cells expressing IFN-γ, read as the spots per well.

### 2.5. Immune Profiling of the Tumor Microenvironment

Tumor tissues were collected after the mice were euthanized (3 mice per treatment condition). Tissue was kept in RNA*later*™ Stabilization Solution (Thermo Fisher, Waltham, MA, USA. AM7020) and were stored at −20 °C. RNA isolation was performed using RNEasy plus Micro kit (Qiagen, Frederick, MD, USA). RNA profiling was performed using the NanoString nCounter Mouse PanCancer Immune Profiling Panel (NanoString Technologies, Seattle, WA, USA). Data analysis was performed using nSolver software (NanoString Technologies).

### 2.6. Data Management and Statistical Analysis

Continuous data were expressed as mean ± standard deviation (SD) or median with 95% CI (confidence interval). Student’s *t*-test and ANOVA were used for comparison analysis. Category data were expressed as percentages and comparative analysis was performed by chi-square test. GraphPad Prism (GraphPad Software version 8.4.3) was used for all statistical analysis, and *p* values less than 0.05 were considered statistically significant.

## 3. Results

### 3.1. Anti-HSV-1 Antibodies Were Successfully Induced after Subcutaneous Inoculation of VG161

To establish anti-HSV-1 immunity, we immunized BALB/c mice by subcutaneous inoculation of mVG161 on days 1 and 7. Anti-HSV-1 antibodies were successfully induced in immunized mice as compared to naïve (non-immunized) mice as shown in Figure 1. The titers of antibodies against HSV-1 in the immunized mice were significantly higher than those of naïve mice (3.3 ± 0.37, mean ± SD) and (0.07 ± 0.01, mean ± SD), respectively (*p* < 0.01) (Figure 1).

### 3.2. Enhanced Anticancer Efficacy in Anti-HSV-1 Antibody-Positive Animals

Next, we tested the inhibitory effect of mVG161 on tumor growth by using two mouse CT26 xenograft models, single and bilateral tumors in immunized and naïve mice. In both models, we used a sub-optimal dose of VG161 which normally would not result in a complete elimination of the tumor from treated mice. The sub-optimal dose and lower frequency (3 consecutive injections instead of 5 injections) was used so that we can discern the differences in the antitumor response between the naïve and immunized mice. In the single-tumor model and in the immunized group, mVG161 significantly inhibited injected tumor growth on days 12, 14, 17 and 19 post treatment as compared to the mVG161-injected tumors in the naïve group. In the naïve groups, the mVG161-injected tumor started growing back on days 14, 17, and 19 after treatment. Comparing the tumor sizes in each group, on day 40, 5 out of 10 mice in the immunized and mVG161-treated group were tumor free, while in naïve mice treated with mVG161, 4 out of 10 mice were tumor free. The survival curve also shows the benefit of immunization in treated mice, where 80% of the mice survived till day 40, compared to only 50% of the naïve group (Figure 2C). Data are shown in Table 1 and Figure 2B.

Next, we tested the inhibitory effect of mVG161 on tumor growth by using mouse CT26 bilateral xenograft models in immunized and naïve mice. In the injected tumor site, mVG161 significantly inhibited injected tumor growth on days 8, 10, 12, 15, 17, and 19 as compared to the injected tumor in the naïve groups, which inhibited tumor growth on days 10, 12, 15, 17, and 19 (Table 2). In the non-injected site, in the naïve groups; none of the tumors were inhibited, but the immunized mice with mVG161 treatment showed the inhibition on days 12, 15, 17, and 19.

Figure 2E shows the mean tumor volume for immunized and naïve mice, and it indicates that mVG161 treatment induced tumor regression in injected and non-injected sites. Moreover, in the immunized groups, the mVG161-injected tumor was completely regressed on day 38 and did not grow back until the end of the study (day 54), indicating a durable response was induced. In the abscopal non-injected tumors, tumor growth was significantly inhibited in immunized and mVG161-treated mice with a delay of 2–3 weeks. Such anticancer activity, however, was not seen in naïve mice. The overall survival for mice from all groups is also illustrated in Figure 2F, and the mice in the immunized and mVG161 treated group survived longer compared to the naïve and mVG161 treated group (Figure 2F).

### 3.3. Enhanced Anticancer Efficacy in Humanized PDX Mouse Model Treated with VG161

The results from CT26 xenograft models indicate that a pre-existing anti-HSV-1 immunity may enhance the anti-tumor activity of VG161 in both injected and abscopal, non-injected tumors. This finding might also support multicycle treatment of VG161, as the first cycle of treatment can induce anti-HSV-1 immunity and thereby enhance the anticancer activity of the following cycle. To verify this hypothesis, we further tested the antitumor activity of VG161 on an ICC PDX model in humanized mice. The engraftment of different immune cells was verified before implanting the tumors (Figure 3A). Two doses of VG161 were tested to examine whether the antitumor effect is dose depend, and five consecutive injections were used per treatment cycle to simulate the currently used treatment regimen in a phase 1 clinical trial [19]. The results showed that multicycle treatment of VG161 significantly inhibited tumor growth (T/C < 40% on day 72). The median time for 3-fold increase in tumor size was 41.5 days (95% CI: 32–58) in the control group, 74 days (95% CI: 51-undefined) in the high dose group, and not reached in the low dose group (*p* < 0.01). There was no significant difference between the two VG161 treatment groups (Figure 3B).

### 3.4. Systemic Specific Anticancer Immunity in Anti-HSV-1 Antibody-Positive Mice

The significant inhibitory effect of mVG161 on non-injected CT26 tumors in immunized mice but not in the naïve mice indicates that the pre-existing anti-HSV-1 immunity may enhance the systemic anticancer immunity induced by mVG161 intratumoral injection and cause the abscopal effect seen in immunized mice. Therefore, we further tested the specific anticancer immunity on lymphocytes isolated from the mouse spleens by ELISPOT assay. As shown in Figure 4, the spot count in the immunized mVG161 treatment group (237.66 ± 79.95) was higher than that in the immunized control group (89.44 ± 28.41), naïve treatment group (135.66 ± 14.39), or naïve control group (98.78 ± 14.40) (mean ± SEM). There was no statistical difference in spot counts among the immunized control group, naïve treatment group, and naïve control group.

### 3.5. Tumor Microenvironment Changes Due to Immunization; NanoString nCounter Analysis

To further investigate possible mechanisms of action for the improvement of treatment efficacy in immunized mice, we used the NanoString nCounter Mouse PanCancer Immune Profiling Panel. This panel examines over 700 genes to identify and track changes to different immune cell types and pathways in the TME. Immune profiling of treated tumors from both immunized and naïve mice revealed a consistently increased score of different immune cells, including cytotoxic cells, NK cells, CD8+ T cells, and CD45 cells (Figure 5A,B). This increase indicates that pre-existing immunity against HSV-1 may enhance tumor infiltration with immune cells following treatment with VG161. Moreover, several pathways related to adaptive immunity have shown higher activation in immunized mice. For example, the functions of antigen-processing cells, leukocytes, and macrophages, as well as MHC expression pathways have shown increased scores in the VG161-injected (treated) tumors from the immunized group compared to the VG161-treated tumors from the naïve group (Figure 5C).

The differential expression of genes revealed that the most significantly upregulated genes in immunized mice were related to T and B cell functions (Lcp1, Lyn, Gbp2b, and Ptprc), cytokines and interferons (Flt3l, Gbp5 and Ifi44), and adhesion molecules (Spn and Sell). On the other hand, genes related to cancer progression (Snai1 and Cdh1) were down regulated in tumors from immunized mice. (Figure 5D).

## 4. Discussion

Pre-existing antivirus immunity in cancer patients has been considered a critical barrier in the clinical development of oncolytic virotherapy, as it may neutralize the OV and thereby significantly reduce tumor cell lysis caused by OVs. Additionally, even in patients without pre-existing antivirus immunity, the anti-drug antibodies induced by the first dosing of OV may deactivate the OV, rendering subsequent dosing ineffective. Therefore, OV has long been considered a “one shot” agent, which further hampers repeated dosing and multicycle treatment with OVs and limits the potential application of OVs in clinical research and practice [20]. Although this might be true for systemically delivered OVs—where the virus is encountered by high titers of neutralizing anti-viral antibodies that render it non-infectious—delivering the OV intratumorally might be different [21]. Neutralizing anti-HSV-1 antibodies can prevent the initial viral infection, if the virus has been previously exposed to the antibodies; however, if the virus is administered intratumorally, the neutralizing antibody will have limited effect on the spread of the viral infection to neighboring cells once the tumor cells have been infected [22,23]. The presence of the neutralizing antibody may not prevent the replication and spread of the virus to other non-infected cells, as the virus may spread via intercellular pathways [22,23,24]. In the current study, the enhancement of anticancer efficacy of HSV-1 OVs was observed in immune-competent tumor-bearing mouse models with pre-existing anti-HSV-1 immunity, which was further confirmed by multicycle treatment with OVs in PDX humanized mice. We have opted to use a sub-optimal dose of the mVG161 treatment in this study, which would not normally result in the complete elimination of the tumors, to discern the difference in the antitumor response between the naïve and immunized mice. Indeed, there was a significant difference in the rate of the tumor growth, with the immunized mice responding better to the OV treatment compared to naïve mice (Figure 2). Similarly, Chahlavi et al. has reported that there was no difference in tumor inhibitory effect of G207, an HSV-1 OV, between HSV-1-seropositive and -seronegative mice, and significantly increased tumor response was observed in mice with multiple treatments as compared to those with a single injection [21]. This result, together with our finding in CT26 xenograft immune-competent mice, and the fact that multi-cycle treatment was proposed in many ongoing clinical trials for oncolytic virus [25], supports repeat dosing and multicycle treatment with HSV-1 OV, which opens the door for clinically using OVs as maintenance therapy. The underlying mechanism for pre-existing antivirus immunity enhancing OV-induced anticancer efficacy is still unclear. Upon intratumoral treatment with HSV-1 OV, we have observed an increase in the number of different types of immune cells, such as T cells and NK cells [11]. In the current work, we have observed similar changes in the tumor microenvironment with increased scores for CD8+ and CD4+ lymphocyte functions, as well as for antigen-presenting cells such as dendritic cells, which were further enhanced in mice with pre-existing anti-HSV-1 immunity. The ELISPOT assay also demonstrated that the antitumor-specific T cells were induced in the spleen after intratumoral injection of OV. These anticancer T cells were higher in mice with pre-existing anti-HSV-1 immunity as compared to naïve mice, although the increase was not statistically significant. There is also evidence indicating that viral antigens may spread from infected cells to other non-infected cells [26]. This may also serve as a mechanism for redirecting anti-viral immunity to attack tumor cells that may have acquired or carry viral antigens, even without being infected. Another explanation for the enhanced efficacy in immunized mice could be attributed to bystander killing effect invoked by the OV treatment [27]. There has been ample evidence that the presence of activated immune cells, along with the inflammatory cytokines produced and secreted by these cells contribute to the killing of the tumor cells [28,29].

Other studies have also reported on the effect of anti-viral immunity on the efficacy of OV treatment. In a recently published phase 1 clinical trial, it was shown that the treatment of gliomas with replication-competent OVs triggers immune responses that rapidly eradicate the therapeutic viruses in most patients, yet induce remarkable and long-lasting antitumor effects in just 20% of the patients [30,31,32]. Pre-existing antiviral immunity has been shown to also enhance adenovirus OV-induced anticancer efficacy [33]. Tähtinen et al. found that the pathogen-related CD4+ T cell memory populations could be re-engaged to support cytotoxic T lymphocytes (CTLs), converting a weak primary antitumor immune response into a stronger secondary one. Treatment with TT-OVA-PeptiCRAd—an oncolytic adenovirus coated with MHC I-restricted tumor-specific peptides and tetanus toxoid (TT) pathogen-specific MHC II-restricted peptides—significantly enhanced antitumor efficacy and induced TT-specific, CD40 ligand-expressing CD4+ T helper cells and maturation of antigen-presenting cells. Moreover, the antitumor effect was even more prominent when combined with the immune checkpoint inhibitor anti-PD-1, strengthening the rationale behind combination therapy with OVs. In line with the findings of increased CD4+ cells in the tumor microenvironment after OV injection, it seems that anti-pathogen CD4+ memory cells play an important role in the enhancement of anticancer activity induced by OVs.

In summary, the present study indicates that pre-existing HSV-1 immunity was associated with increased anticancer efficacy and increased infiltration of immune cells in tumors treated with VG161

One of the limitations of this study is that although we could demonstrate increased infiltration of different immune cells (including T-cells) in the tumors from immunized mice, we were not able to identify the specificity of these cells (whether they were anti-tumor or anti-viral). This work could be the target of another investigation.

As there is always a translational gap from the preclinical setting to the clinic, the proposed strategy for antitumor efficacy enhancement needs to be verified in patients by appropriately designed clinical trials. For example, the interval between the injections is critical for achieving the best efficacy and should be determined by monitoring the neutralizing antibody level induced by pre-immunization in patients. The common interval seen in ongoing clinicals is about 2–6 weeks, which need to be optimized by measuring the neutralizing antibody level after first dosing. In addition, although the underlying mechanism was preliminarily explored, it remains to be further elucidated by well-designed preclinical and clinical studies.

## 5. Conclusions

The anticancer efficacy of VG161 can be enhanced by pre-immunization with HSV-1 and multicycle administration when the virus is given intratumorally, indicating that pre-existing antiviral immunity might enhance OV-induced antitumor immunity. Our results suggest potential clinical benefits of HSV-1-based OV therapy in HSV-1-seropositive patients and multicycle administration of VG161 for long-term maintenance treatment. Clinical trials in HSV-1-seropositive patients are ongoing.

## Figures and Tables

**Figure 1 viruses-14-02327-f001:**
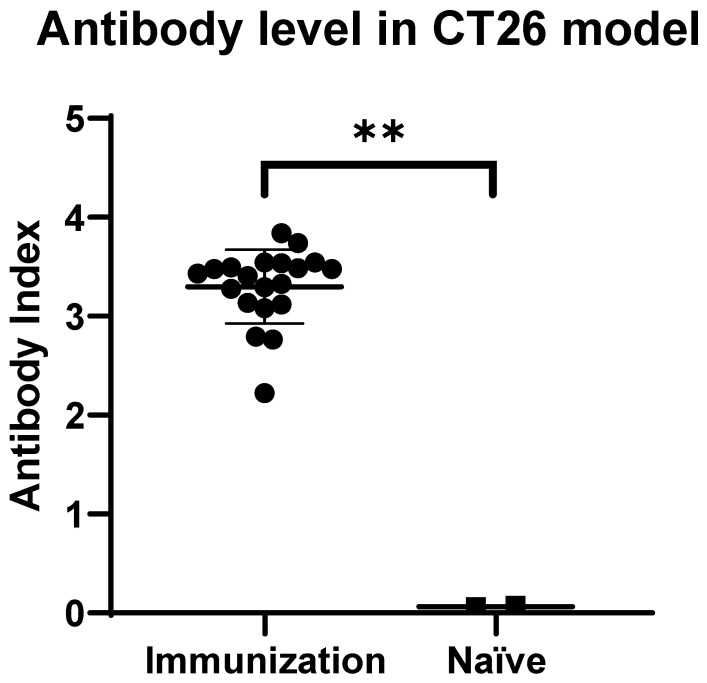
Anti-HSV-1 response after immunization in CT26 mouse models: Before the tumor implantation, serum was randomly collected from 2 naïve mice and 20 immunized mice. The anti-HSV-1 antibody level was detected using Mouse/Rat HSV-1 IgG ELISA kit. ** *p* < 0.05.

**Figure 2 viruses-14-02327-f002:**
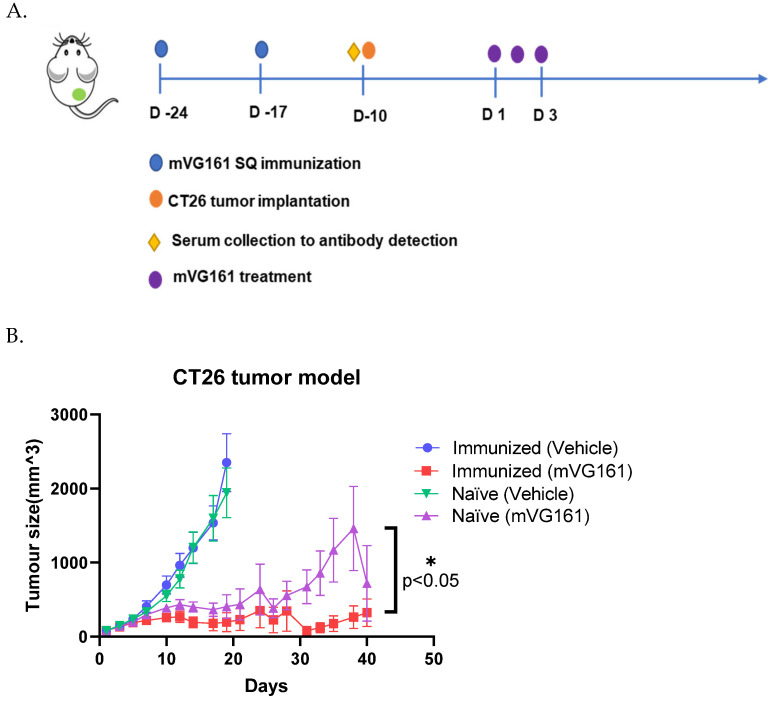
Efficacy and survival of CT26 mouse model treated with mVG161. (**A**) The schema of treatment protocol for Balb/c mice used in the CT26 tumor mode, (**B**) Mean tumor volume in each group in mice treated with mVG161 (single tumor model) (**C**) Survival curve for mice treated with mVG161 or vehicle, either immunized or naïve mice (single tumor model). (**D**) Immunization and treatment regimen for Balb/c mice used in the CT26 bilateral tumor model (**E**) Mean tumor volume in each group in mice treated with mVG161 (bilateral tumor model). (**F**) Survival of mice treated with mVG161 (Bilateral tumor model).

**Figure 3 viruses-14-02327-f003:**
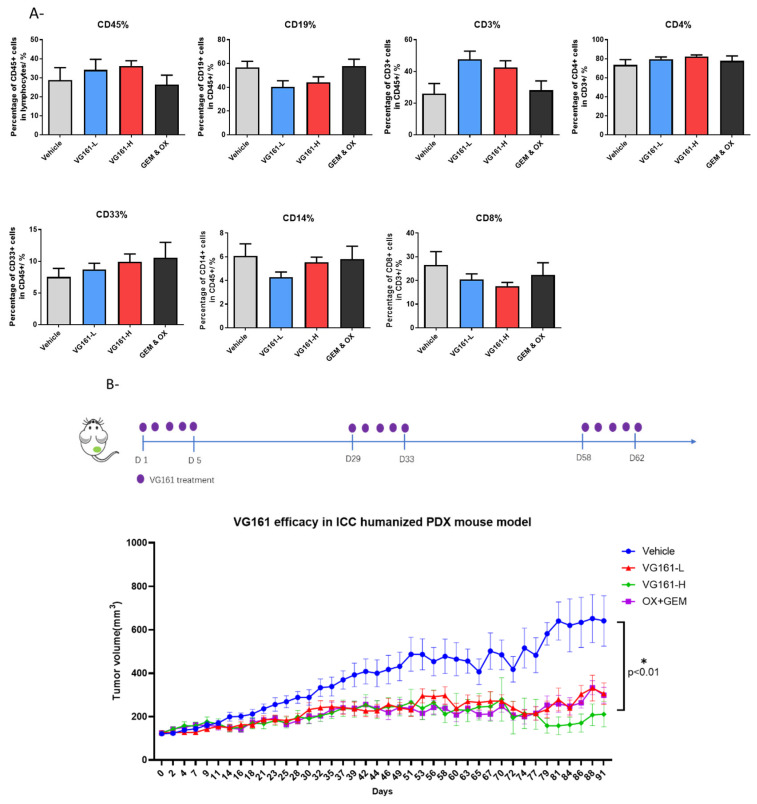
Humanized ICC PDX model. (**A**) Implantation of different human white blood cells in the different mouse groups before treatment. (**B**) Tumor size after administering VG161 to female Hu-NPI mice bearing patient tumor tissue. Data points represent group mean; error bars represent standard error of the mean (SEM).

**Figure 4 viruses-14-02327-f004:**
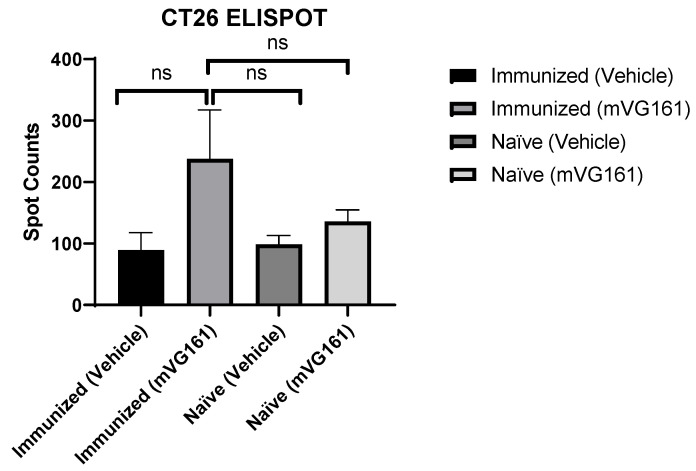
CT26 Balb/c mouse models. Splenocytes were collected from each mouse 10 days after administering mVG161 or vehicle control. Cells were co-incubated with CT26 cells. ELISPOT kit was used to detect IFN-γ expression. Data points represent number of spots for each mouse. Error bars represent standard error of the mean (SEM).

**Figure 5 viruses-14-02327-f005:**
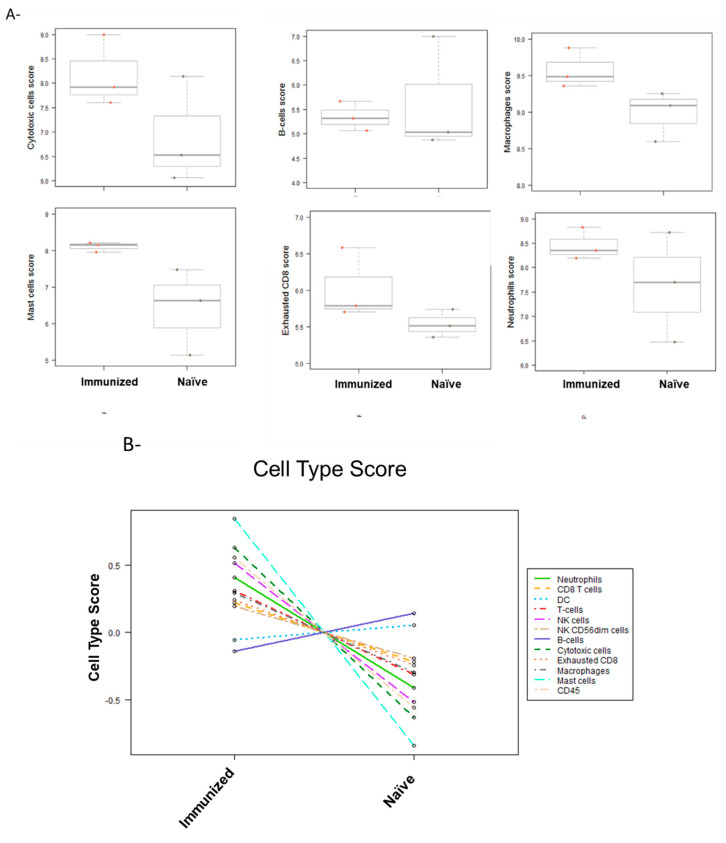
Cell scores in tumors treated with mVG161 comparing immunized vs. naïve mice (**A**,**B**). Pathway analysis shows the most significant changes in different pathways when the same tumors were analyzed (**C**). differential gene analysis of the most significantly up or down-regulated genes in the tumors of naïve and mVG161 treated mice compared to immunized and mVG161 treated mice in CT26 tumors. (**D**) Volcano plot of differentially expressed genes in tumors from naïve treated mice compared to immunized treated mice. This analysis was performed using the NanoString platform.

**Table 1 viruses-14-02327-t001:** Summary of mean ratio of T/C in single-tumor model.

Group	D1	D3	D5	D7	D10	D12	D14	D17	D19
**Immunized (mVG161)**	100.00%	100.59%	87.30%	59.33%	41.88%	32.52%	19.72%	14.79%	10.83%
**Naïve (mVG161)**	100.00%	94.53%	83.44%	89.42%	69.04%	55.02%	33.41%	22.89%	20.01%

**Table 2 viruses-14-02327-t002:** Summary of mean ratio of T/C in bilateral-tumor model.

Day		D1	D3	D5	D8	D10	D12	D15	D17	D19
Mean T/C% ^1^										
Injected Side	immunized (mVG161)	100.0%	72.1%	52.2%	37.0%	27.9%	25.2%	22.9%	28.7%	28.3%
	naïve (mVG161)	100.0%	98.8%	73.2%	50.8%	39.1%	32.5%	32.1%	48.2%	58.3%
Non-injected Side	immunized (mVG161)	100.0%	112.0%	112.4%	81.1%	54.9%	39.6%	36.2%	45.2%	44.2%
	naïve (mVG161)	100.0%	96.0%	103.1%	78.7%	65.4%	58.7%	62.4%	81.2%	84.1%

^1^ Note: (1) T/C % was calculated using the following formula: T/C% = 100% × RTV treatment/RTV control (RTV = V_t_/V_0_, V_t_: Tumor volume at the day of measurement. V_0_: Tumor volume before the virus treatment. (2) On Day 22, all the mice in the vehicle group were euthanized due to the tumor size.

## Data Availability

The data presented in this study are available on request from the corresponding author.

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
