# Peer review of "Pre-Existing HSV-1 Immunity Enhances Anticancer Efficacy of a Novel Immune-Stimulating Oncolytic Virus"

_viruses, 2022, doi:10.3390/v14112327_

Round 1

Reviewer 1 Report

Dear authors,

Please see attached review letter for comments. 

Author Response

In this study, authors investigated the impact of preexisted immunity against HSV-1 on anticancer efficacy of VG161, an HSV-1 oncolytic virus. They provided convincing evidence of enhanced anticancer efficacy of intertumoral injection of oncolytic virus by preimmunization with the virus. The findings challenge the dogma that preexisted immunity to oncolytic virus may compromise anticancer activity. The question is whether multicycles of OV delivery is indeed better than one-shot, which should be discussed in detail. Any suggestion to the interval between each injection? The following are minor comments:

Q: The question is whether multicycles of OV delivery is indeed better than one-shot, which should be discussed in detail.

A: We have further discussed this in the discussion and results sections, and also referenced several papers, including a current clinical trial for VG161, where up to 5 injections of the OV is injected per treatment cycle.

Q: Any suggestion to the interval between each injection?

A: “The interval between the injections is critical for achieve the best efficacy and should be determined by monitoring the neutralizing antibody level induced by pre-immunization in patients. The common interval seen in ongoing clinicals is about 2-6 weeks, which need to be optimized by measuring the neutralizing antibody level after first dosing.” This has also been added to the discussion part of the manuscript.

  • The immunization protocol should be clarified in the MM. The first immunization was on day 1, boost on day 7. It was not clear when the blood was collected, on the same day (day 7) of the boost, or 7 days after the boost. If it was 7 days post the boost, it should be labeled as day 14 (page 3, line 99-100, 110).

We have adjusted the description in the methods’ section and generated 3 panels (Figures 2a, 2d, and 3b) to better illustrate the treatment schedule.

  1. The Authors stated that “The titers of antibodies against HSV-1 in both control (3.3 ± 0.37, mean

± SD) and treatment (0.07 ± 0.01, mean ± SD) groups of immunized mice were significantly

higher than those of non-immunized mice (p<0.01) (Page 4, line 168, and Figure 1).” It looks like

it was mislabeled. The “control” should be the treated, or immunized group, please check and

correct. Please make the terms consistent, control, naïve, treated, immunized, vehicle, etc. to

avoid misunderstanding.

The text has been corrected, and we made sure to make the terms used consistent through out the manuscript (immunized/naïve).

  1. Figure titles should be in the figure ligands, not separately in front of each figure.

The figures have been adjusted to include the legends.

  1. Figure 2a, naïve (mVG161). The tumor volume dropped suddenly on day 40. Please give a

reasonable explanation. The same with figure 2C (right, solid square group), unlikely that the

tumor volume changed from around 800 mm3 to unmeasurable in a few days (2-3 days?).

the reason for the drop in the average volume was due to some mice in the control groups reaching the end point due to large tumor volume, so they had to be euthanized. This is evident from the survival curve (figure 2c). The same happened for the other groups. When mice needed to be euthanized, the average will drop. We’ve also added a survival curve (2f) to indicate when mice were euthanized.

  1. Figure 2c is too small, hard to read.

Figures have been remade and colors were used to make it easier to distinguish between different groups.

  1. Figure 5a is too small too. Any information regarding Treg cells, an important

immunosuppressive population?

Unfortunately, nanostring analysis didn’t show Tregs as significantly impacted by the pre-immunization.  We’ve included another panel to discuss the differentially expressed genes, which we felt may also be relevant. Also, we’ve added few points to the discussion section that we felt may further address the immune response, along with possible explanation of the enhanced efficacy we’ve seen in immunized mice (e.g., bystander effect).

  1. In figure 4 legend, authors showed that Error bars represent standard error of the mean (SEM),

but in the manuscript described as mean ± SD. Please check and verify consistence.

The correct legend should be SEM, which has been changed in the text. 

  1. Statistics, please label every figure with statistic significance, p value and other parameters as

needed.

Statistics were performed and p values were added to the figures where statistical significance was shown.

  1. Table 2 showed unusual values on day 22 compared to that on day 19, which does not seem to

be consistent with figure 2C. Please check, verify, and discuss. Please include the actual value

(tumor size or volume) of treated group and control group, which may give a clearer picture to

the readers.

Yes, the numbers on day 22 jumped significantly, and the reason is that many of the control mice had to be euthanized at that point due to increased tumor burden (and tumor size), resulting in much higher ratios. We have deleted the points past day 19 for this reason. We’ve also added the survival curve (figure 2F) as a supplement to figure 2E, which shows the average tumors sizes of each group. We hope this addresses the concern.

Reviewer 2 Report

Ding et al. investigated the effects of pre-existing HSV-1 immunity on the anti-tumoral efficacy of a novel oncolytic HSV (VG161). As we know, pre-existing immunity is one of the barriers in front of oncolytic virotherapy and this considers a hot topic in this field. While the study design is appropriate, there are some key points missing in this manuscript, and the following concerns need to be addressed.

Major comments:

11)      The authors could be evaluated spleen cells to differentiate anti-viral and anti-tumoral responses (as they mentioned in the discussion) and mechanisms behind tumor regression following pre-existing immunity. The authors state that “it seems that anti-pathogen CD4+ memory cells play an important role in the enhancement of anticancer activity induced by OVs” but there isn’t any data in support of their statement.

22)      All the graphs need to be revised, and significancy between the groups should be inserted. Moreover, P values should be inserted in the results section.

Minor comments:

11)      In line 401, the authors discussed figure 5C but they didn't show the figure. Also, the legend needs to be revised.  

22)      In the results section (nanostring nCounter analysis, line 388): the authors should be explained in detail the transcriptomic variation and highlight the role of effective genes. 

33)      Figure 3A, the resolution of this figure is not clear.

44)      Figures caption should be placed in below the figures (Fig. 2 and 3), please consider this issue. 

55)      In page 8, line 280; the authors mentioned briefly about results of two VG161-treated groups (VG161-H: high dose and VG161-L: low dose): What was the logic? The authors should be explained clearly in the results and discussion section. It seems that the low doses have shown better results than high doses (line 282).  

66)      Plenty of studies have shown intratumoral injection of oncolytic viruses leads to regression of tumor volume, although in the naive mVG161 group (Fig. 2C) it seems that there is no significancy, especially on the 20th day onwards. I'd like to know the interpretation of the authors in this regard. 

Author Response

Reviewer #2 Questions-

Ding et al. investigated the effects of pre-existing HSV-1 immunity on the anti-tumoral efficacy of a novel oncolytic HSV (VG161). As we know, pre-existing immunity is one of the barriers in front of oncolytic virotherapy and this considers a hot topic in this field. While the study design is appropriate, there are some key points missing in this manuscript, and the following concerns need to be addressed.

Major comments:

1)      The authors could be evaluated spleen cells to differentiate anti-viral and anti-tumoral responses (as they mentioned in the discussion) and mechanisms behind tumor regression following pre-existing immunity. The authors state that “it seems that anti-pathogen CD4+ memory cells play an important role in the enhancement of anticancer activity induced by OVs” but there isn’t any data in support of their statement.

As this is a critical point in this work (to show the contribution of the anti-viral response to the tumor regression), and we’ve demonstrated the presence of the anti-viral immune response by testing the antibody response against the virus, where a quantitative commercial kit is available. Although we’re not showing in this study, work that has been conducted by other groups show the indirect, “bystander” effect of immune cells on fighting tumors. We have discussed and cited some of this work in the discussion section. 

Demonstrating the contribution of such response in the tumor microenvironment and differentiating the anti-viral vs anti-tumor response proved more challenging. We have added a section to the discussion (study limitations) to highlight this point.

2)      All the graphs need to be revised, and significancy between the groups should be inserted. Moreover, P values should be inserted in the results section.

All figures have been revised and updated to address this comment.

 Minor comments:

  • In line 401, the authors discussed figure 5C but they didn't show the figure. Also, the legend needs to be revised.

The label for figure 5C was deleted due to formatting issue, however, that should be the heatmap. Label has been fixed.  

  • In the results section (nanostring nCounter analysis, line 388): the authors should be explained in detail the transcriptomic variation and highlight the role of effective genes. 

We have expanded this section to elaborate more on the different finding from the nanostring analysis. We have also added a volcano plot depicting differentially expressed genes to the results section. The contribution of the major cells present was discussed, as well as the role of the most highly differentially expressed genes.

  • Figure 3A, the resolution of this figure is not clear.

The figures have been remade to improve the visibility and quality.

  • Figures caption should be placed in below the figures (Fig. 2 and 3), please consider this issue. 

The location of the figures’ captions have been adjusted to address this concern.

  • In page 8, line 280; the authors mentioned briefly about results of two VG161-treated groups (VG161-H: high dose and VG161-L: low dose): What was the logic? The authors should be explained clearly in the results and discussion section. It seems that the low doses have shown better results than high doses (line 282).

The logic for setting the 2 dose levels is to test whether the antitumor efficacy is dose dependent. Clearly, there was no dose dependency seen, as there is no significant difference between the 2-treatment group.

We have also reworded the results section to address the concern about the multi-cycle treatment. We’ve also referenced the current clinical trail for VG161, where up to 5 injections of the drug is used per treatment cycle to explain the rational for using the 5 injections per cycle regimen in this experiment.

6)      Plenty of studies have shown intratumoral injection of oncolytic viruses leads to regression of tumor volume, although in the naive mVG161 group (Fig. 2C) it seems that there is no significancy, especially on the 20th day onwards. I'd like to know the interpretation of the authors in this regard. 

You are right. In fact, we have shown that 5 IT injections of VG161 would result in tumor regression and complete elimination of the tumor (please check our cited published paper, reference #11). However, for this study, we opted to use a much lower dose and treatment frequency to elucidate the effect of pre-immunization on the treatment.

We have added to the results and the discussion that a “suboptimal dose” was used to elucidate the effect of pre-existing immunity on the treatment efficacy.

Reviewer 3 Report

The authors have investigated the role of pre-existing immunity on therapeutic outcome following HSV in CT26 model and a human PDX model. They demonstrate that immunised mice are more able to control the tumour growth of both injected tumour and more interestingly non-injected tumours. This finding is of clinical relevance and also expands on our current knowledge in the OV field of the role of anti-viral immunity and successful therapy.

The paper would be improved if statistical analysis was included on the graph and exact p number written in the text. Moreover on lines 24-25 a re-wording of tumour lymphocytes specific to tumour specific lymphocytes. Again on line 273- Inhibitory effects to anti-cancer activity.

Figure 3 should include a schematic of viral and chemotherapy treatment. In the text the authors say they hypothesise that multi-treatment schedule would be more beneficial in a PDX model, but the comparison between single vs multi treatment is not investigated and therefore this needs to be reworded or even better the experiment performed so those conclusions can be made.

Author Response

Reviewer #3 Questions-

The authors have investigated the role of pre-existing immunity on therapeutic outcome following HSV in CT26 model and a human PDX model. They demonstrate that immunised mice are more able to control the tumour growth of both injected tumour and more interestingly non-injected tumours. This finding is of clinical relevance and also expands on our current knowledge in the OV field of the role of anti-viral immunity and successful therapy.

The paper would be improved if statistical analysis was included on the graph and exact p number written in the text.

We have calculated and included statistical analysis and provided the p-value where a significant difference was observed in the results.

Moreover on lines 24-25 a re-wording of tumour lymphocytes specific to tumour specific lymphocytes.

The statement was changed

 Again on line 273- Inhibitory effects to anti-cancer activity.

The statement was changed

Figure 3 should include a schematic of viral and chemotherapy treatment. In the text the authors say they hypothesise that multi-treatment schedule would be more beneficial in a PDX model, but the comparison between single vs multi treatment is not investigated and therefore this needs to be reworded or even better the experiment performed so those conclusions can be made.

Schematics have been added to figures 2 and 3 to illustrate the treatment schema for each experiment.

The logic for setting the 2 dose levels is to test whether the antitumor efficacy is dose dependent. Clearly, there was no dose dependency seen, as there is no significant difference between the 2-treatment group.

We have also reworded the results section to address the concern about the multi-cycle treatment. We’ve also referenced the current clinical trail for VG161, where up to 5 injections of the drug is used per treatment cycle to explain the rational for using the 5 injections per cycle regimen in this experiment.

Round 2

Reviewer 2 Report

The authors improved their manuscript. It could be considered for publication in the viruses journal.